# Dynamics of the Degradation of Acetyl-CoA Carboxylase Herbicides in Vegetables

**DOI:** 10.3390/foods10020405

**Published:** 2021-02-12

**Authors:** Miroslav Jursík, Kateřina Hamouzová, Jana Hajšlová

**Affiliations:** 1Faculty of Agrobiology, Food and Natural Resources, Czech University of Life Sciences Prague, 165 00 Prague, Czech Republic; hamouzova@af.czu.cz; 2Department of Food Analysis and Nutrition, University of Chemistry and Technology Prague, 166 28 Prague, Czech Republic; jana.hajslova@vscht.cz

**Keywords:** herbicide residues, non-residual production, low-residual production, pesticide degradation in vegetables

## Abstract

(1) Background: Aryloxyphenoxy-propionates and cyclohexanediones are herbicides most widely used in dicot crops worldwide. The main objective of the study was to determine the dynamics of herbicide residues in carrot, lettuce, cauliflower, and onion in order to suggest a low level of residues in harvested vegetables. (2) Methods: Small plot field trials were carried out in four vegetables in the Czech Republic. The samples of vegetables were collected continuously during the growing season. Multiresidue methods for the determination of herbicide residues by LC-MS/MS were used. Non-linear models of degradation of individual herbicides in vegetables were calculated using the exponential decay formula. Action GAP pre-harvest intervals for the 25% and 50% maximum residue limit (MRL) and 10 µg kg^−1^ limit (baby food) were established for all tested herbicides. (3) Results: The degradation dynamics of fluazifop in carrot, onion, and cauliflower was significantly slower compared to quizalofop and haloxyfop. The highest amount (2796 µg kg^−1^) of fluazifop residues was detected in cauliflower 11 days after application. No residue of propaquizafop and cycloxydim was detected in any vegetable samples. (4) Conclusions: Aryloxyphenoxy-propionate herbicide (except propaquizafop) could contaminate vegetables easily, especially vegetables with a short growing season. Vegetables treated with fluazifop are not suitable for baby food. Lettuce and cauliflower treated by quizalofop are not suitable for baby food, but in onion and carrot, quizalofop could be used. Propaquizafop and cycloxydim are prospective herbicides for non-residual (baby food) vegetable production.

## 1. Introduction

Aryloxyphenoxy-propionates (FOPs) and cyclohexanediones (DIMs) are the most commonly used leaf graminicides (herbicides against grass weeds) in dicotyledonous (dicot) crops [1,2,3]. Herbicides of both chemical groups block the conversion of acetyl-CoA to malonyl-CoA by inhibiting the activity of the enzyme acetyl-CoA carboxylase (ACCase). This inhibition of fatty acid synthesis blocks the production of phospholipids used in building new cell membranes required for cell growth [4]. Dicot plants are naturally tolerant of these herbicides because of an insensitive ACCase, but tank-mix combinations with other herbicides could cause great damage to treated crops, especially under unsuitable weather conditions [5].

FOPs are formulated and applied as an ester of their acids. Emulsifiable concentrate is a common formulation for these herbicides. After application, esters are rapidly converted to acids by carboxyesterase activity. Acid forms are readily translocated to the meristematic tissues through the phloem. This process inhibits the growth of young developing leaves of susceptible grass weeds [6]. Necroses of the growing points are visible two weeks after application depending on the temperature.

FOPs are applied post-emergently because they are taken up by the leaves of plant, but some herbicide will also fall to the soil [7]. The soil fate of FOPs will depend on soil pH, with slower degradation in alkaline soils [8]. There is also a concern of water being contamination with these herbicides, as documented in Greek surface waters with quizalofop [9] and Brazilian rivers with haloxyfop [10]. On the other hand, Mantzos et al. [11] showed minimal risk of the contamination of the soil and adjacent water by quizalofop. These herbicides had transient, harmful effects on most of the soil’s microbiological parameters [12].

Conventional farming practices include the use of pesticides within an integrated pest management program for crop protection against diseases, pests, and weeds. However, pesticides are potentially toxic to humans and can have both acute and chronic health effects, depending on the quantity and ways in which a person is exposed [13]. For this reason, it is necessary to model pesticide distribution in harvested crops [14]. Maximum residue limits (MRLs) for pesticides in crops were established by the European Union in a regulation of the European Commission [15] as the highest level of pesticide residues that are tolerated in food. Low-residual production is a designation for agricultural production where residues of used pesticides in harvested products are below the limit for a predetermined action threshold, for example 25% MRL or 50% MRL [13,16]. Furthermore, non-residue production is a classification for products with pesticide residues below the limit of 10 µg kg^−1^. This limit is currently used worldwide for baby foods [17]. Some FOPs (diclofop, haloxyfop, quizalofop) contain chlorine, and these herbicides could metabolize to chlorate, especially in carrots and potatoes. These would be unacceptable for baby food products [18].

Accumulation of pesticide residues in vegetables is less frequent than in fruits [19]. Within vegetables, pesticide residues are most widely detected in pepper and cucumber, while samples of lettuce, cauliflower, and carrots seldom contained pesticide residues [20,21]. This is not to say that residues are never detected in these vegetables. Elgueta et al. [22] and Skovgaard et al. [23] detected pesticide residues in 50% of lettuce samples with 16–20% of samples above the MRL. Santareli et al. [24] also detected many lettuce samples over the MRL in Italy. Even organically managed vegetables could be occasionally contaminated by pesticide residues, especially in countries where the control of pesticide use is less strict. For example, in Brazil, a large number of organic carrot samples contain pesticide residues [25].

Most of above-mentioned studies detected insecticides or fungicides in vegetables, but few studies have monitored the contamination of vegetables by herbicides. For example, Sing et al. [26] monitored pre-emergent herbicides in carrots, but no residue was detected. Similar results published by Saritha et al. [27] did not detect any residues of metribuzin in tomato. Khan et al. [28] detected residues of linuron in onion, carrot, and lettuce. There are a few cases where pendimethalin was detected in vegetables [29,30], especially in vegetables with a short growing season like lettuce [31] or kohlrabi [32]. Risk of contamination by FOP residues is quite high because late post-emergence application is on the label, but to our knowledge, no study has focused on the degradation dynamics of FOPs yet.

The main objective of this study was to determine the degradation dynamics of herbicide residues in carrot, lettuce, cauliflower, and onion to provide suggestions for low-residue production of harvested vegetables. The specific objectives of this work were (1) to quantify the concentration of herbicide residues in tested vegetables, (2) to develop recommendations for herbicide weed control for low-residual and non-residual vegetable production, and (3) to recommend the safest leaf graminicide for each tested vegetable.

## 2. Materials and Methods

Small plot field trials were carried out in carrot (variety Grivola), onion (variety Wellington), lettuce (variety Elenas), and cauliflower (variety Chamborg) in a field of the University of Life Sciences, Prague, the Czech Republic (300 m a.s.l., 50°7′ N, 14°22′ E) in 2012–2016. The region has a temperate climate with an annual mean air temperature of about 9 °C and a mean annual precipitation of about 500 mm. None of the tested herbicides were used in previous crops (potatoes). Each vegetable was grown in a separate growing area with specific agrotechnological requirements (soil preparation, fertilization, and irrigation). Common agricultural practices of the European and Mediterranean Plant Protection Organization were used. Experimental plots were arranged in randomized blocks. Plot size was 16 m^2^ (2 m × 8 m) for each vegetable. Crop density, row spacing, and planting/sowing times are given in Table 1.

The samples of vegetable (roots of carrot, bulbs of onion, leaves of lettuce and florets of cauliflower) were collected continuously during the growing season from the central part of each plot. There was a two-week interval between the first and second sampling and between the second and third sampling. A minimum of four plants was collected from one plot during each sampling term. The samples were stored at −20 °C until the extraction procedure.

All tested herbicides (Table 2) were formulated as emulsifiable concentrates. A small-plot sprayer, fitted with a Lurmark 015F110 nozzle, was used to apply the herbicides. The application pressure was 0.25 MPa, and the water volume applied was 300 L/ha. The maximum registered rates of all tested herbicides were used. Herbicides were applied post-emergently in two terms (Table 3).

Analyses of pesticide residues were performed by the testing laboratory of the University of Chemistry and Technology using the LC-MS/MS method accredited according to the EN ISO/IEC 17025 standard [33]. The analytical method used in this study is based on EN standards [34].

In brief, the following steps were performed: (i) alkaline hydrolysis (10 mL of acetonitrile and 2 mL of 5 M NaOH added to 10 g of homogenized sample, shaking 2 h at 40 °C); (ii) acidification (2 mL of 2.5 M H_2_SO_4_) and addition of 100 μL formic acid); (iii) QuEChERS like extraction (addition of 4 g MgSO_4_ and 1 g of NaCl; and internal standard, triphenylphosphate, then, intensive shaking; centrifugation to separate acetonitrile phase for further analysis). An aliquot of the upper organic layer was transferred to a vial for LC-MS/MS. An Acquity UPLC HSS T3 analytical column (100 mm × 2.1 mm, 1.8 μm particle size, Waters, USA) and mobile phases consisting of (A) water with 5 mM ammonium formate/0.1% (*v*/*v*) formic acid and (B) methanol were used for ultra-high performance liquid chromatography (U-HPLC) in extract separation. A triple quadrupole mass spectrometer (Xevo TQ-S, Waters, Milford, MA, USA) with electrospray ionization in a positive ion mode (ESI+) was used for the final identification and quantification of herbicide residues (Table 4). The method used for residues analysis was fully validated in line with the requirements stated in the European Commission’s guidance document SANTE/12682/2019 [35]. Limits of quantification together with maximum residue limits (MRLs) established by Regulation EC 396/2005, are summarized in Table 5. The extended uncertainty of measurement at 0.01 mg/kg level was 15%. To avoid results bias due to matrix effects, matrix-matched calibration was used.

The generated data was processed using MassLynx software version 4.1 (Waters Corporation, Milford, USA). External quality control was ensured by regular participation in proficiency tests of the European Commission’s Proficiency Testing Program.

The obtained data were processed in R project version 3.6.1 (R Core Team, 2019) and subjected to the comparison analysis (*t*-test) to reflect the differences in experimental years. Non-linear models of degradation of individual herbicides in crops were calculated using the exponential decay formula in drc package using the following equation:y = a(exp(−x/b)) (1)
where y was the amount of active ingredient (µg kg^−1^), x was number of days after herbicide application, parameter b > 0 determined the steepness of decay, and a was the upper limit. Goodness of fit was assessed by F-test. All tests were performed using a significance level of 0.05. Parameters of models and the analytical results are shown in Table 6.

Action pre-harvest intervals for the 50% MRL (APHI_50_), 25% MRL (APHI_25_), and 10 µg kg^−1^ limit (APHI_BF_) were established for fluazifop and quizalofop in all four vegetables and for haloxyfop in onion and carrot. The above-mentioned equation was used to calculate the APHI (as variable x). For non-residue production, a concentration of 10 µg kg^−1^ was used. A given percentage of the MRL (50 and 25%) was calculated for low-residue production. The calculated value of APHIs was extended by one-third depending on a confidence interval of the model for each herbicide with the aim of increasing the reliability of APHIs, i.e.,
APHI = t + (1/3t).(2)

The MRL parameter differs according to the active ingredient of the herbicide and the vegetable. The PHI was used when the computed APHI value extended by one-third was lower than the compulsory PHI as indicated in the list of registered products [36].

## 3. Results

### 3.1. Carrot

The degradation dynamics of fluazifop in carrot were significantly slower compared to quizalofop and haloxyfop. Relatively high differences in the degradation dynamics of fluazifop were recorded among the experimental growing season and application term, nevertheless, the differences were not statistically significant. The highest concentrations of fluazifop (above 500 µg kg^−1^) in carrot were detected in the first two weeks after application. The degradation dynamics of quizalofop and haloxyfop were similar and any analyzed samples of onion did not contain more than 150 µg kg^−1^ (Figure 1). For fluazifop, quizalofop, and haloxyfop, the values of APHI_50_ were shorter than PHI. Similarly, the values of APHI_25_ for quizalofop and haloxyfop (25 and 45 days, respectively) were shorter than PHI (45 and 56 days, respectively). In such cases, the APHI was not relevant. While modeling the quizalofop curve, the negative value appeared. It was due to low observed values in the experiment and high MRL. The longest APHI_BF_ (110 days) was calculated for fluazifop (Table 7). Any residue of propaquizafop and cycloxydim was not detected in any sample of carrot, regardless of the application term and year of sampling.

### 3.2. Onion

Fluazifop showed the slowest degradation in onion, especially during the first four weeks after application. The highest concentration of fluazifop (235 µg kg^−1^) in onion was detected 7 days after application. The degradation dynamics of quizalofop and haloxyfop was similar and all analyzed samples of onion contained no more than 70 µg kg^−1^ (Figure 2). For quizalofop, the calculated APHI_25_ was equal to the APHI_BF_ due to a low MRL (40 µg kg^−1^). For fluazifop and haloxyfop, the APHI_50_ and APHI_25_ were shorter than PHI. Modeled values of haloxyfop reached a negative value (Table 6); this is due to two facts. First, the MRL established by authorities is quite high and, second, the observed values did not reach such a high value. For quizalofop, the APHI_BF_ (29 days) was shorter than PHI (42 days). In such cases, the APHI was not relevant (Table 7). Incidence of propaquizafop residue was not detected in any tested onion sample. Cycloxydim was not tested in onion.

### 3.3. Lettuce

During the first two weeks after application, fluazifop showed slower degradation in lettuce compared to quizalofop (Figure 3). The degradation dynamics of fluazifop were affected by the lettuce head size at the time of application. The highest concentration of fluazifop (350–550 µg kg^−1^) in lettuce was detected when herbicide was applied on almost ripened heads. In contrast, lettuce head size had no effect on quizalofop degradation. For fluazifop, the calculated APHI_50_ and APHI_25_ were equal and longer than APHI_BF_ (60 days) due to a low MRL (20 µg kg^−1^). For quizalofop, the APHI_50_ was not relevant because it is shorter than PHI. APHI_BF_ for quizalofop was 71 days (Table 7). Residues of propaquizafop and cycloxydim were not detected in any tested lettuce sample. Haloxyfop was not tested in lettuce.

### 3.4. Cauliflower

The degradation dynamics of fluazifop in cauliflower were significantly slower compared to quizalofop in all growing seasons (Figure 4). The concentration of fluazifop in all analyzed cauliflower samples was 18–220 times higher than MRL (10 µg kg^−1^). In contrast, the concentration of quizalofop in cauliflower did not exceed 100 µg kg^−1^ in any tested sample. For fluazifop, the calculated APHI_50_ and APHI_25_ were not relevant due to a low MRL; APHI was 105 days (Table 7). For quizalofop, 18 (50% MRL), 33 (25% MRL), and 68 days (for baby food) APHIs were calculated, while the PHI is prescribed as 70 days for brassica vegetables. No residues of propaquizafop and cycloxydim were found in any tested cauliflower sample. Haloxyfop was not tested for in cauliflower.

Days needed to reach the hypothetically set-up MRL values to 10, 20, 50, 100, 200, and 500 µg kg^1^, were calculated (Table 8). The aim here was to establish the baseline for a case in which authorities re-establish MRLs at new levels. Out of three active ingredients tested, the longest time was observed in fluazifop, regardless of the vegetable. In most cases, the model crops used in this study will obtain an MRL equal to 100 µg kg^−1^.

## 4. Discussion

Of the tested leaf graminicides, **fluazifop** exhibited the slowest degradation dynamics in all tested vegetables. The highest amount (2796 µg kg^−1^) of fluazifop residues was detected in cauliflower 11 days after application. Similar degradation dynamics of fluazifop were recorded by Doohan et al. in strawberry [37]. In their study, residues of fluazifop ranged between 50 and 3240 µg kg^−1^ within 12–28 days after application. The half-life of fluazifop in vegetable leaf (lettuce and spinach) was relatively low and ranged from 1.11 to 2.27 days [38]. Balinova and Lalova detected no residues of fluazifop in soybean seeds after harvest [39]; however, Sondhia detected a relatively high concentration of fluazifop both in straw (472–702 µg kg^−1^) and seeds (297–312 µg kg^−1^) of soybean after post-emergence application [40]. Risk of contamination of cauliflower and lettuce by fluazifop residue is relatively high due to the low MRL (10, resp. 20 µg kg^−1^) and slow dissipation. Moreover, the growing period of these vegetables is short (less than 60 days for lettuce and less than 90 days for cauliflower) and leaf graminicides are usually used 3–5 weeks after planting crops. The MRL for fluazifop in onion and carrot is considerably higher (300, resp. 400 µg kg^−1^), therefore, fluazifop could be used in these vegetables, along with canopies, for low-residual production (up to 25% MRL). Vegetables treated by fluazifop are not suitable for baby food due to the long APHI (53–110 days).

Concentration of **quizalofop** residues in all tested vegetable samples were below the MRL and did not exceed 400 µg kg^−1^ in lettuce and 100 µg kg^−1^ in carrot, onion, and cauliflower. Relatively low quizalofop residues in groundnut plants (104 µg kg^−1^ 30 days after application) were detected by Poonia et al. [41]. In their study, quizalofop residues decreased below detection limit (10 µg kg^−1^) 60 days after application. In blueberry fruit, no residues were detected two weeks after split application of quizalofop [42]. The half-life of quizalofop in potato leaves ranged from 0.04 to 13.1 days in study of Wang et al., and no residues were detected in leaves and tubers at harvest [7]. Similar results were presented by Mantzos et al., who detected quizalofop residues in stems and leaves of sunflower 18 days after application, but no residues in inflorescences and seeds at harvest time [11]. In addition, in our study, the degradation dynamics of quizalofop was slower in leaves of lettuce compared to roots of carrot, tubers of onion, or florets of cauliflower. This theory confirms the study of Sahoo et al., where they reported a very short half-life of quizalofop in onion (0.85 day) and no residue was detected at harvest time [43]. The risk of contamination of tested vegetables by quizalofop residues is low because of the fast degradation dynamics. The calculated APHI for 25% MRL did not exceed PHI in any tested vegetable. Lettuce and cauliflower treated by quizalofop are not suitable for baby food due to long APHI (60, resp. 105 days), while onion and carrot could be treated by quizalofop (APHI for baby food 42, resp. 55 days).

A remarkably low quantity of residue was detected after application of **haloxyfop**. Only one sample of carrot (119 µg kg^−1^ 4 days after application) exceeded the MRL. The degradation dynamics of haloxyfop in onion was fast and no onion samples contained more than 60 µg kg^−1^. However, concentration of haloxyfop residues in onion leaves could be significantly higher (100 and 800 µg kg^−1^ 10 days after application) [44]. Our suggested APHIs for 25% MRL did not exceed PHIs in onion and carrot. APHIs for baby food in these vegetables were relatively long (47 and 64 days, respectively), but possible.

No residues of **propaquizafop** were detected in any tested vegetable sample. Moreover, Duhan and Sing did not detect any residues of propaquizafop (detection limit 3 µg kg^−1^) in cotton seeds and lint at harvest time [45]. No other relevant studies about the degradation dynamics of propaquizafop in vegetables have been published yet, such as the degradation dynamics of cycloxydim. Both herbicides seem to be prospective for non-residual (baby food) vegetable production.

In comparison to other herbicides in vegetables, only residues of the soil active herbicide pendimethalin have frequently contaminated lettuce [31]. Contamination of cauliflower by herbicide clomazone, clopyralid, picloram, quinmerac, metazachlor, pyridate, dimethachlor, dimethenamid-P, S-metolachlor, napropamide, and pendimethalin was not detected in the study of Suk et al. [46]. This herbicide is usually applied shortly before or after planting/sowing and, therefore, a longer PHI could be achieved. Applications of leaf graminicides were carried out later, resulting in a PHI that was actually shorter. The main result of this study is that aryloxyphenoxy-propionate herbicides (except propaquizafop) could contaminate vegetables easily, especially vegetables with a short growing season.

## 5. Conclusions

Aryloxyphenoxy-propionate herbicide (except propaquizafop) could contaminate vegetables easily, especially vegetables with a short growing season. Vegetables treated with fluazifop are not suitable for baby food. Lettuce and cauliflower treated by quizalofop are not suitable for baby food, but in onion and carrot, quizalofop could be used. Propaquizafop and cycloxydim are prospective herbicides for non-residual (baby food) vegetable production.

## Figures and Tables

**Figure 1 foods-10-00405-f001:**
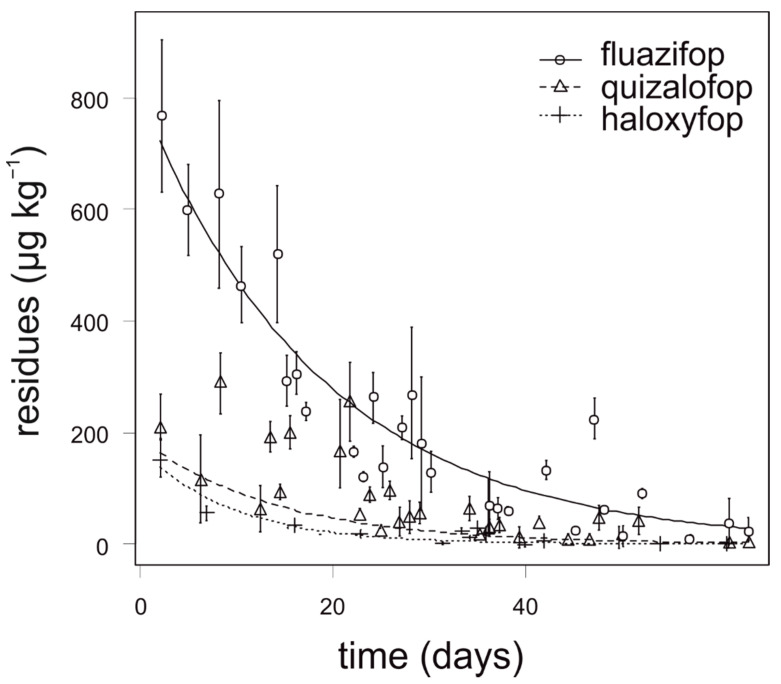
Degradation dynamics of fluazifop, quizalofop, and haloxyfop in carrot. Mean values with error bars representing standard error of the mean.

**Figure 2 foods-10-00405-f002:**
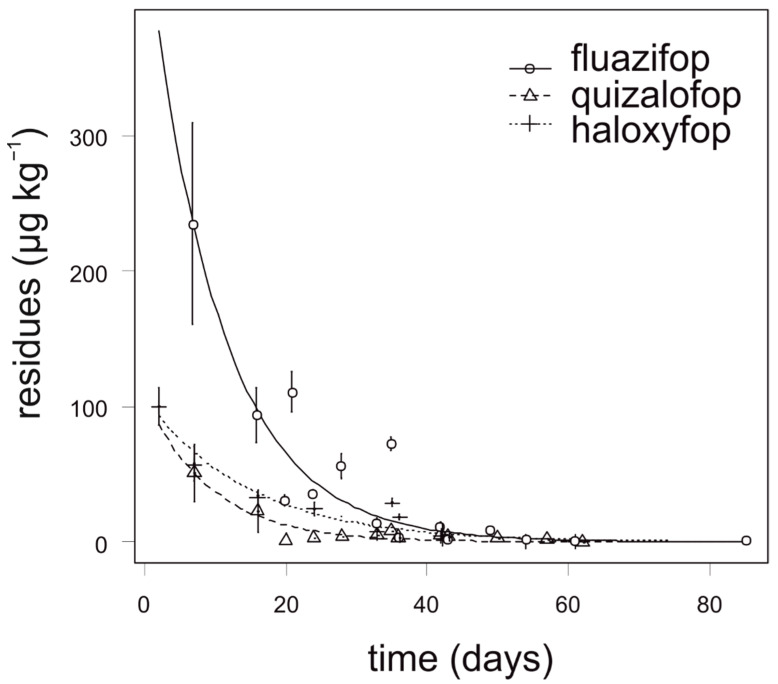
Degradation dynamics of fluazifop, quizalofop, and haloxyfop in onion. Mean values with error bars representing standard error of the mean.

**Figure 3 foods-10-00405-f003:**
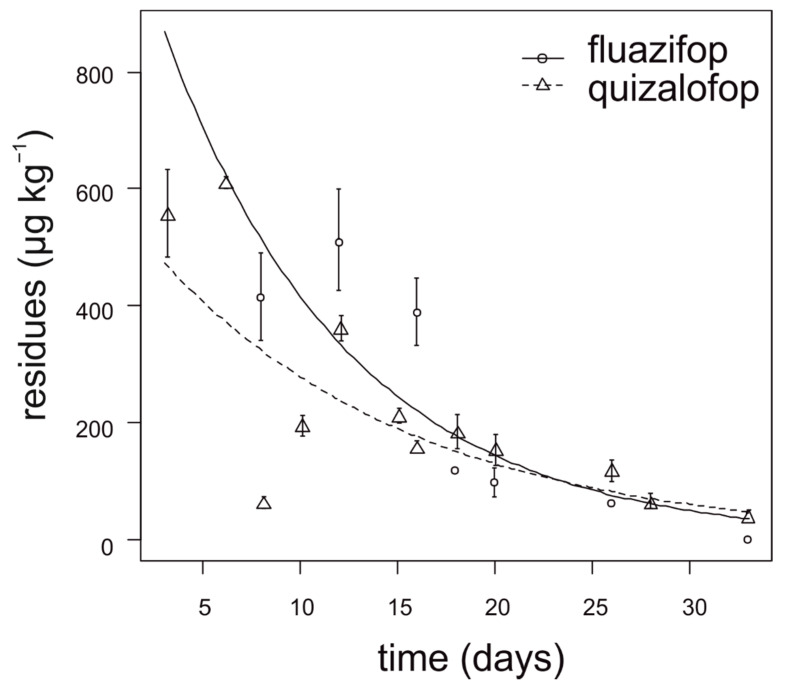
Degradation dynamics of fluazifop and quizalofop in lettuce. Mean values with error bars representing standard error of the mean.

**Figure 4 foods-10-00405-f004:**
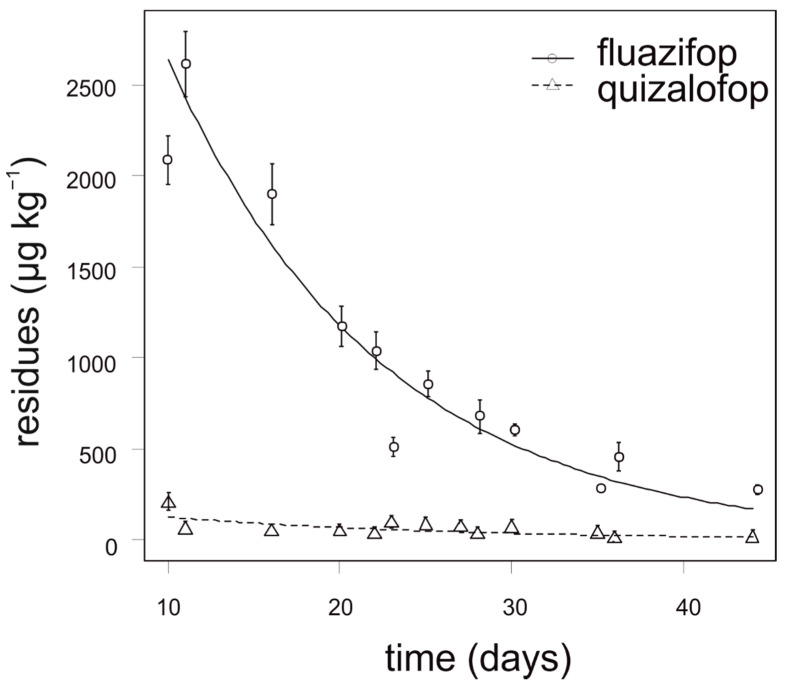
Degradation dynamics of fluazifop and quizalofop in cauliflower. Mean values with error bars representing standard error of the mean.

**Table 1 foods-10-00405-t001:** Crop and plot arrangements.

Vegetable	Crop Density (Plant m^−2^)	Inter-Row Spacing (m)	Date of Planting/Sowing				
			2012	2013	2014	2015	2016
carrot	90	0.5	25.3	20.4	31.3	21.4	28.4
onion	70	0.3	29.3	15.4	12.3	30.3	-
lettuce	10	0.3	28.3	9.4	27.3	-	-
cauliflower	4	0.5	9.5	6.5	5.5	-	-

**Table 2 foods-10-00405-t002:** Description of tested herbicides.

Active Ingredient (ai)	Trade Name	Concentration of ai (g L^−1^)	Application Rate (g ha^−1^ ai)	Manufacturer
cycloxydim	Stratos Ultra	100	200	BASF
fluazifop	Fusilade Forte	150	300	Syngenta
haloxyfop	Gallant Super	104	104	Corteva
propaquizafop	Agil	100	150	ADAMA
quizalofop	Targa Super	100	250	Chemtura

**Table 3 foods-10-00405-t003:** Term of herbicide application in experimental years.

Vegetable	Growth Stage	Date of Herbicide Application				
		2012	2013	2014	2015	2016
carrot	5 TL ^1^	27.6	24.6	16.6	17.6	16.6
	7 TL	9.7	7.7	23.6	30.6	28.6
	9 TL	-	15.7	7.7	-	-
onion	6 L ^2^	11.6	7.7	11.6	17.6.	-
	9 L	27.6	22.7	7.7	13.7.	-
lettuce	4 WAP ^3^	15.5	13.5	23.5	-	-
	6 WAP	25.5	29.5	3.6	-	-
cauliflower	6 WAP	19.6	24.6	23.6	-	-
	8 WAP	27.6	7.7	7.7	-	-

^1^ true leaves, ^2^ leaves, ^3^ weeks after planting.

**Table 4 foods-10-00405-t004:** Mass spectrometric detector setting.

Analyte	Quantification	Cone	Collision	Confirmation	Cone	Collision
	Transition (*m*/*z*)	(V)	(V)	Transition (*m*/*z*)	(V)	(V)
Cycloxydim	326.3 > 280.2	30	13	326.3 > 180.4	30	25
Fluazifop	328.2 > 282.1	35	20	328.2 > 91.2	35	30
Haloxyfop	362 > 315.8	27	18	362 > 91	27	30
Propaquizafop	444.2 > 100.04	30	20	444.2 > 56.2	30	15
Quizalofop	344.46 > 298.83	54	18	346.46 > 300.83	54	18

**Table 5 foods-10-00405-t005:** Limits of quantification and maximum residue limit for tested herbicides and vegetables.

Herbicide	Carrot		Onion		Lettuce		Cauliflower	
	LOQ ^1^	MRL ^2^	LOQ	MRL	LOQ	MRL	LOQ	MRL
	µg kg^−1^							
cycloxydim	2	5000	-	3000	2	1500	2	5000
fluazifop	2	400	1	300	1	20	1	10
haloxyfop	2	90	2	200	-	10	-	10
propaquizafop	2	200	2	40	2	400	2	200
quizalofop	2	200	1	40	1	400	1	200

^1^ limit of quantification, ^2^ maximum residue limit.

**Table 6 foods-10-00405-t006:** Parameters of the exponential decay model and analytical results.

Vegetable	Active Ingredient	Parameter					
		a ^1^	SE ^2^	b ^1^	SE	F-Test ^3^	*p*-Value
carrot	fluazifop	804.09	121.91	18.82	3.39	0.55	0.91
	quizalofop	78.56	13.19	32.89	7.85	1.67	0.16
	haloxyfop	153.15	19.57	17.66	2.43	0.94	0.64
onion	fluazifop	458.81	67.95	10.30	1.18	2.28	0.12
	quizalofop	108.76	25.03	9.24	1.73	0.40	0.90
	haloxyfop	81.98	12.08	18.57	2.94	7.57	0.13
lettuce	fluazifop	1194.64	617.99	9.45	3.77	8.97	0.30
	quizalofop	262.89	120.05	29.10	21.72	3.91	0.14
cauliflower	fluazifop	5905.47	800.93	12.4	1.31	7.77	0.12
	quizalofop	107.99	33.25	26.77	10.17	4.95	0.18

^1^ parameters of model - *a* represents upper limit of the curve, *b* is the steepness of the decay, ^2^ standard error, ^3^ significance *p* = 0.05.

**Table 7 foods-10-00405-t007:** Active pre-harvest interval for tested herbicides in tested vegetables for current MRL.

Vegetable	Active Ingredient	MRL ^1^ (µg kg^−1^)	Model (µg kg^−1^)	PHI ^2^ (Days)	APHI_BF_ ^3^ (Days)	APHI_25_ ^4^ (Days)	APHI_50_ ^4^ (Days)
carrot	fluazifop	400	13.14	49	110	52	35
	quizalofop	200	−0.81 ^5^	45	55	25	12
	haloxyfop	90	9.38	56	64	45	29
onion	fluazifop	300	4.38	28	53	25	15
	quizalofop	40	9.24	42	29	29	21
	haloxyfop	200	−9.29 ^5^	28	47	15	2
lettuce	fluazifop	20	38.64	42	60	69	60
	quizalofop	400	5.22	30	71	31	19
cauliflower	fluazifop	10	79.13	56	105	128 ^5^	117 ^5^
	quizalofop	200	2.57	70	68	33	18

^1^ maximum residue limit, ^2^ pre-harvest interval, ^3^ active pre-harvest interval for baby food (10 µg kg^−1^), ^4^ active pre-harvest interval for low-residual production (25, resp. 50% MRL, ^5^ value is less than active pre-harvest interval for baby food (10 µg kg^1^) due to low MRL, ^5^ Due to the low observed values in experiment and high MRL value, the model results in some negative predictions for quizalofop (carrot) and haloxyfop (onion).

**Table 8 foods-10-00405-t008:** Hypothetical pre-harvest interval for different MRL.

Vegetable	Active Ingredient	MRL 10 ^1^ (Days)	MRL 20 ^2^ (Days)	MRL 50 ^3^ (Days)	MRL 100 ^4^ (Days)	MRL 200 ^5^ (Days)	MRL 500 ^6^ (Days)
carrot	fluazifop	110	70	57	52	26	9
	quizalofop	55	45	15	12	x	x
	haloxyfop	64	36	20	10	x	x
onion	fluazifop	53	32	23	21	9	x
	quizalofop	29	16	7	1	x	x
	haloxyfop	47	26	9	1.26	x	x
lettuce	fluazifop	60	39	34	31	17	8
	quizalofop	71	65	48	31	8	x
cauliflower	fluazifop	105	83	75	67	42	31
	quizalofop	68	45	21	18	x	x

^1^ maximum residue limit 10 µg kg^−1^; ^2^ 20 µg kg^−1^; ^3^ 50 µg kg^−1^; ^4^ 100 µg kg^−1^; ^5^ 200 µg kg^−1^; ^6^ 500 µg kg^−1^. In some cases, the calculated parameters reached value < 0, this is not meaningful from practical point of view, so these are marked as x.

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
