# Peer review of "Dynamics of the Degradation of Acetyl-CoA Carboxylase Herbicides in Vegetables"

_foods, 2021, doi:10.3390/foods10020405_

Round 1

Reviewer 1 Report

The submitted manuscript entitled “Aryloxyphenoxy-propionates: herbicides with low dynamic of degradation in vegetables” is in a line with the current research trend regarding the potential of toxic activity of pesticides, which covering both potential acute and chronic health effects to humans.

Authors studied the dynamics of herbicide residues, e.g. fluazifop, quizalofop, haloxyfop in carrots, lettuce, cauliflower and onions. The presented topic is interesting from a scientific and practical point of view, especially due to the poorly understood model of pesticide distribution in the harvested crops.

In my oppinion, although Conclusion section is not obligatory, it would be advisable to include a few summary sentences at the end of the article.

Furthermore, Figure 1 should be moved the Section 3.1. Carrot and Figure 3 to Section 3.3. Lettuce.

Author Response

I would like to thank reviewer for her/his comments.

I added short Conclusions paragraph to chapter Discussion. I agree with reviewer: Figure 1 should be moved to the section 3.1. Carrot and Figure 3 to the section 3.3. Lettuce.

Reviewer 2 Report

Title:
Aryloxyphenoxy-propionates: herbicides with low dynamic of 2 degradation in vegetables
please change article title to be more informative
Dynamics of degradation ...or Degradation of ... The degradation dynamics

l119 - Sentence:
standard (Foods of plant origin-Multiresidue methods for the determination of pesticide residues by GC or LC-MS/MS-Part 2: Methods for extraction and clean-up) and EN 12393-2 (Foods of plant origin-Multiresidue methods for the determination of pesticide residues by GC or LC-MS/MS-Part 3: Determination and confirmatory tests).
please move to reference list.

Please describe in detail analytical system LC-MS, column type, mobile phases, reagents, etc.

Figures - please add SD data to the result points

Author Response

Point 1: Please change article title to be more informative
Dynamics of degradation ...or Degradation of ... The degradation dynamics

Response 1: The Title of the manuscript was changed: Dynamic of the degradation of ACCase herbicides in vegetables

Point 2: L119 - Sentence: standard (Foods of plant origin-Multiresidue methods for the determination of pesticide residues by GC or LC-MS/MS-Part 2: Methods for extraction and clean-up) and EN 12393-2 (Foods of plant origin-Multiresidue methods for the determination of pesticide residues by GC or LC-MS/MS-Part 3: Determination and confirmatory tests).
please move to reference list.

Response 2: This EN standard was added to the reference list (No.34).

Point 3: Please describe in detail analytical system LC-MS, column type, mobile phases, reagents, etc.

Response 3: Analytical methodology was specify

Point 4: Figures - please add SD data to the result points

Response 4: Standard deviation and mean values are now depicted in figures 1-4.

Reviewer 3 Report

The present research studies the dynamic of degradation of 4 aryoxyphenoxy-propionates and one cyclohexanedione herbicides in different vegetables in order to assess their use suitability in baby food vegetables production. Although the research is not highly novel, the results can be considered of interest. However, several problems/doubts should be solved:

  • Taking into account that a cyclohexanedione was included in the study, the title of the manuscript is not appropriate. This should be replaced by “Dynamic of the degradation of ACCase herbicides in vegetables”.
  • Line 128-132. The QuEChERS method should be provided.
  • Line 132-135. The UPLC-MS/MS method should be provided, including selected mass parameters (collision energy, precursor ions, product ions, …).
  • Line 135-138. The validation data should be provided in the table 4 (LOD, LOQ, Linearity, RSD, recovery, Matrix effect…).
  • Table 4. “DL” should be replaced by “LOD”.
  • Line 147-153. A mathematical equation with chemical meaning should be used to describe the degradation kinetics of the herbicides. Likewise, it should be indicated why this equation was selected and not another.
  • Table 5. Regarding to the parameters obtained in cauliflower, the authors should discuss why the degradation kinetics differ significantly from those previously obtained and published by these authors in the years 2012-2014 (https://doi.org/10.17221/312/2018-PSE). This issue should be included in section 4.
  • Figures 1-4. Barr errors should be depicted.
  • Tables 4, 6, 7. Standard errors should be provided.

Author Response

Point 1: Taking into account that a cyclohexanedione was included in the study, the title of the manuscript is not appropriate. This should be replaced by “Dynamic of the degradation of ACCase herbicides in vegetables”.

Response 1: The Title of the manuscript was changed: Dynamic of the degradation of ACCase herbicides in vegetables

Point 2: Line 128-132. The QuEChERS method should be provided.

Response 2: The QuEChERS method was provided: In brief, following steps were performed: (i) alkaline hydrolysis (10 ml of acetonitrile and 2 ml of 5M NaOH added to 10 g of homogenized sample, shaking 2 hours at 40°C); (ii) acidification (2 ml of 2,5M H2SO4) and addition of 100 μl formic acid); (iii) QueChers like extraction (addition of 4 g MgSO4 and 1 g of NaCl; and internal standard, triphenylphosphate,  then, intensive shaking; centrifugation to separate acetonitrile phase for further analysis).

Point 3: Line 132-135. The UPLC-MS/MS method should be provided, including selected mass parameters (collision energy, precursor ions, product ions, …).

Response 3: The UPLC-MS/MS method was provided: An Acquity UPLC HSS T3 analytical column (100 mm × 2.1 mm, 1.8 μm particle size, Waters, USA) and mobile phases consisting of (A) water with 5 mM ammonium formate / 0.1% (v/v) formic acid and (B) methanol were used for Ultra-High Performance Liquid Chromatography (U-HPLC) was used for extract separataion. A triple quadrupole mass spectrometer (Xevo TQ-S, Waters, Milford, MA, USA) with electrospray ionization in a positive ion mode (ESI+), was used for the final identification and quantification of herbicide residues (Table 4). The method used for residues analysis was fully validated, in line with the requirements stated in the European Commission’s guidance document SANTE/12682/2019. Limits of quantification together with maximum residue limits, MRLs established by Regulation EC 396/2005 are summarized in Table 5, the extended uncertainty of measurement at 0.01 mg/kg level was 15%. To avoid results bias due to matrix effects, matrix-matched calibration was used.

Point 4: Line 135-138. The validation data should be provided in the table 4 (LOD, LOQ, Linearity, RSD, recovery, Matrix effect…).

Response 4: The validation data was provided in new table (Table 4)

Point 5: Table 4. “DL” should be replaced by “LOD”.

Response 5: DL was replaced by LOQ

Point 6: Line 147-153. A mathematical equation with chemical meaning should be used to describe the degradation kinetics of the herbicides. Likewise, it should be indicated why this equation was selected and not another.

Response 6: Commonly used equation of herbicide degradation processes are usually the single first-order kinetics, bi-phasic kinetics and Lag-phase models (eg. Noshadi et Homaee, 2018, https://doi.org/10.1016/j.still.2018.06.005). Also, the equation like mean half-life (DT50) and time required for degradation of 90% herbicide (DT90) values are often calculated.

Generally, statisticians recommend two basic approchases for mathematical modelling of the data 1) use the same equation as was used previously in a published article, 2) fitt the data to the best model, you can find. In this case, we decided for the second approach. Nevertheless, the equation used in this study is exponential one and it was found as the best fitting function in R. Similar function with two parameters to that used in other studies describing degradation of pesticides or herbicides.

Point 7: Table 5. Regarding to the parameters obtained in cauliflower, the authors should discuss why the degradation kinetics differ significantly from those previously obtained and published by these authors in the years 2012-2014 (https://doi.org/10.17221/312/2018-PSE). This issue should be included in section 4.

Response 7: Less data were calculated in previous paper. In this manuscript, more data was included and therefore the estimated values have changed. Honestly, we think that the model fit is improved, due to more values in dataset.

Point 8: Figures 1-4. Barr errors should be depicted.

Response 8: Standard deviation and mean values are now depicted in figures 1-4.

Point 9: Tables 4, 6, 7. Standard errors should be provided.

Response 9: In table 4 (5), there are values established by Regulation EC 396/2005. These values were not calculated from the data obtained in experiments, therefore, standard errors cannot be provided. Standard errors in table 6 (7) and 7 (8) are not provided because calculated APHIs value were extended by one-third depending on a confidence interval of the model for each herbicide. The same method of qualify of APHIs was used in previous paper in this journal: Horská, T.; Kocourek, F.; Stará, J.; Holý, K.; Mráz, P.; Krátký, F.; Kocourek, V.; Hajšlová, H. Evaluation of pesticide residue Dynamics in lettuce, onion, leak, carrot and parsley. Foods 2020, 9, 680.
